# Solid-Contact Ion-Selective Electrodes for Histamine Determination

**DOI:** 10.3390/s21196658

**Published:** 2021-10-07

**Authors:** Siyuan Ma, You Wang, Wei Zhang, Ye Wang, Guang Li

**Affiliations:** State Key Laboratory of Industrial Control Technology, College of Control Science and Engineering, Zhejiang University, Hangzhou 310027, China; 11732017@zju.edu.cn (S.M.); king_wy@zju.edu.cn (Y.W.); 0619595@zju.edu.cn (W.Z.); 22032093@zju.edu.cn (Y.W.)

**Keywords:** solid-contact, histamine electrode, porphyrins, artificial cerebrospinal fluid

## Abstract

Solid-contact ion-selective electrodes for histamine (HA) determination were fabricated and studied. Gold wire (0.5 mm diameter) was coated with poly(3,4-ethlenedioxythiophene) doped with poly(styrenesulfonate) (PEDOT:PSS) as a solid conductive layer. The polyvinyl chloride matrix embedded with 5,10,15,20-tetraphenyl(porphyrinato)iron(iii) chloride as an ionophore, 2-nitrophenyloctyl ether as a plasticizer and potassium tetrakis(p-chlorophenyl) borate as an ion exchanger was used to cover the PEDOT:PSS layer as a selective membrane. The characteristics of the HA electrodes were also investigated. The detection limit of 8.58 × 10^−6^ M, the fast response time of less than 5 s, the good reproducibility, the long-term stability and the selectivity in the presence of common interferences in biological fluids were satisfactory. The electrode also performed stably in the pH range of 7–8 and the temperature range of 35–41 °C. Additionally, the recovery rate of 99.7% in artificial cerebrospinal fluid showed the potential for the electrode to be used in biological applications.

## 1. Introduction

Histamine (HA) is synthesized through the oxidative decarboxylation of the amino acid histidine. It acts as an inflammatory mediator in inflammatory and allergic reactions [1]. Controlled by immune signals, mast cells in peripheral connective tissues store and release HA to deal with antigen exposure and pathological conditions such as tissue injury, autoimmunity and inflammation [2,3]. In human skin suffering from urticaria, the content of HA in pigmentosa is ten times more than that in normal skin tissue [4]. As a signaling molecule in pathophysiological and physiological processes, HA is also involved in gastric acid secretion, tumor genesis and vasodilation [5]. In tumors extremely rich in HA, the unprecedented value of nearly 1000 μg HA per gram of tissue is much higher than the highest value of HA (200–280 μg/g tissue) in any normal tissue [6]. Moreover, HA functions as a modulatory neurotransmitter in cognition and the regulation of sleep behavior in the mammalian brain. The dysfunction of the histaminergic system in the central nervous system has been proved to be related to insomnia, Parkinson’s disease and addictive behaviors [7,8]. HA is not only an endogenous substance in vertebrates, but an indicator of food freshness. In fish causing food poisoning, the more than 50 mg of HA per 100 g of fish is orders of magnitude greater than the 0.01 mg/100 g of HA in fresh fish [9,10]. Thus, developing sensitive and fast HA determination methods is a growing public health concern worldwide. It is necessary for disease diagnosis and food-safety supervision.

In recent years, various HA determination methods have been researched, including high-performance liquid chromatography (HPLC) [11,12], thin-layer chromatography (TLC) [13], colorimetry [14], fluorescence methods [15,16], and enzyme-linked immunosorbent assay (ELISA) [17]. Although some methods are of good sensitivity and specificity, they are restricted by their high costs, their time-consuming properties, the need for derivation and pretreatment due to the complexity of the sample matrix, and the low analytical concentrations of HA. In comparison with these methods, electrochemical analyses have been rapidly developed in recent years due to their advantages of low cost, fast responses, reproducibility, reasonable sensitivity and storage stability [18,19,20]. When amperometric electrodes, one of the electrochemical analytical methods, are utilized for HA determination, a high potential is necessary for a redox reaction with which to detect the current characteristics, and the redox product is unfavorable for the nervous system [21,22]. Although modern impedimetric electrodes, another electrochemical method, minimize unwanted cell depolarization with lower voltages [23], the complex models and analyses are not universal [24].

By contrast, potentiometric determination with ion-selective electrodes (ISEs) has many more advantages. For instance, no operating potential and no enzyme are required, they allow easy storage, and they have relatively long life spans of at least one month [25,26]. In contrast to conventional HA electrodes, which are internally filled with electrolyte solution, the solid conductive layer in solid-contact ISEs transduces ionic signals into electronic conductivity. Thus, they are easier to maintain and miniaturize [27]. Compared with other conductive polymers such as polyaniline and polypyrrole, one of the most widely used commercial products, poly(3,4-ethlenedioxythiophene) doped with poly(styrenesulfonate) (PEDOT:PSS), offers the advantages of good conductivity and less sensitivity to O_2_/CO_2_, light and temperature [28,29]. To obtain sensitivity to HA, different ionophores have been used in the polymeric selective membrane. However, *α*-cyclodextrin shows poor selectivity for inorganic ions widely distributed in the human body [30]. Dibenzo-30-crown-10 shows no response to monocationic HA, which leads to the invalidation of HA sensors when the pH values of solutions are higher than 6 [31]. Compared with these ionophores, metalloporphyrins bond more stably with aromatic amines than aliphatic amines. This leads to a stronger interaction between porphyrins and HA in the membrane [32].

In this work, solid-contact ISEs for HA determination were studied. A PEDOT:PSS layer was coated on a 0.5 mm-diameter gold wire to take the place of the electrolyte solution in conventional electrodes. The HA selective membrane contained 5,10,15,20-tetraphenyl(porphyrinato)iron(iii) chloride (Fe(TPP)Cl) as an ionophore, 2-nitrophenyloctyl ether (NPOE) as a plasticizer and potassium tetrakis(p-chlorophenyl) borate (KTFPB) as an ion exchanger. The characteristics of the HA sensors were investigated, and the electrodes showed good sensitivity to HA, selectivity for common interferences, stability at different pHs and temperatures, and excellent reproducibility. The long-term stability over one month was also found to be acceptable. In order to test the possibility of HA determination in biological applications, artificial cerebrospinal fluid was used as the background in the standard addition method.

## 2. Experimental

### 2.1. Reagents

Fe(TPP)Cl, NPOE, high-molecular-weight poly(vinyl chloride) (PVC), KTFPB, the aqueous dispersion of PEDOT:PSS (1.3 wt% dispersion in H_2_O), HA, D-(+)-glucose (GC), dopamine hydrochloride (DA), urea (Ur), epinephrine hydrochloride (EP), and L-ascorbic acid (AA) were purchased from Sigma Aldrich (Shanghai, China). Lithium hydroxide monohydrate (LiOH), acetate dehydrate (CH_3_COOH), potassium chloride (KCl), sodium chloride (NaCl), calcium chloride (CaCl_2_), hydrochloric acid (HCl), hexadecyltrimethylammonium bromide (CTAB) and tetrahydrofuran (THF) were purchased from Sinopharm Chemical Reagent (Shanghai, China). D-histidine (HI) was obtained from Macklin Reagent (Shanghai, China). A 0.5 mm-diameter gold wire with a 99.99% metal basis was acquired from Alfa Aesar (Shanghai, China). The artificial cerebrospinal fluid was purchased from Sinopharm Chemical Reagent (Shanghai, China). Purification was not needed for any of the chemicals.

The HA stock solution (10^−1^ M) was prepared by dissolving 0.555 g of HA in 50 mL of acetic buffer (CH_3_COOH-LiOH). Other HA solutions with concentrations of 10^−2^–10^−7^ M were obtained by diluting a 10^−1^ M HA stock solution in acetic buffers. All the solutions were stored at 4 °C.

### 2.2. Fabrication of the HA ISE

The gold wire with 7 cm length was inserted into a capillary tube and sealed with resin. One end of the gold wire was exposed as the conductive substrate (the active area was approximately 0.0942 cm_2_) and polished with alumina. After it had been rinsed with deionized water, ethanol and sulfuric acid were applied for ultrasonic cleaning, successively. Then, the gold wire was dried with nitrogen and stored in a drying closet.

The aqueous dispersion of conductive polymer was prepared by mixing 0.1 M CTAB (0.1 vol%) as the surfactant in PEDOT:PSS (99.9 vol%), and the dispersion was shaken homogeneously with a vortex. Then, a 6 mm length of the gold wire was immersed into the dispersion and taken out to dry out. After this step was repeated 18 times, the gold wire was entirely covered with a conductive layer and put upside down in the drying closet for 12 h.

The HA selective membrane was prepared by mixing Fe(TPP)Cl as the ionophore, KTFPB as the ion exchanger and PVC as the membrane matrix in a tube. Once the THF, as the solvent, and NPOE, as the plasticizer, had been added to the tube, the mixture was stirred immediately for better dissolution. After the membrane cocktail was homogeneous and had been stored at 4 °C for 24 h, the selective membrane was coated on the gold wire by dipping as described previously [33]. The structure and photography of the HA sensor are shown in Figure 1A,B, respectively. In Figure 1C, it can be observed that the surface of the membrane was uniform without defects. The homogeneously distributed white plots referred to as ionophores enhanced the interaction with HA and improved the sensitivity.

### 2.3. Evaluation of the Potentiometric Response

Potentiometric measurements were carried out with CHI660 Electrochemical Work Station (Shanghai Chenhua Inc., Shanghai, China). A Ag/AgCl single-junction electrode containing 3 M KCl as the external filling solution worked as the reference electrode and formed a two-electrode system with the working electrode. The pH values were determined by using a glass-pH electrode (Mettler Toledo, Shanghai, China), and a shaking water bath (Shanghai Lichen Inc., Shanghai, China) was used to test the influence of temperature. All the experiments were conducted at 25 °C and a pH value of 7.4, at which the monoprotonated form of HA is dominant.

The selectivity coefficients for several secondary ions were measured and calculated by the fixed interference ion method (FIM) using the constant concentration of respective chlorides [34]. In order to determine the recovery rate for HA in real samples, the standard addition method was applied by adding a high concentration of HA solution into the artificial cerebrospinal fluid. Before every measurement, the electrode was conditioned by soaking the membrane in 1 mM HA solution for a period of 1 h before use.

## 3. Results

### 3.1. Influences of Membrane Composition and Thickness of the Membrane

Four types of membrane cocktails were prepared by mixing the PVC, plasticizer, ionophore and ion exchanger in proportion. After being covered with the membrane, HA electrodes with different ionophore and ion exchanger contents were tested as summarized in Table 1. The performance of these electrodes was evaluated according to the IUPAC recommendations [34]. The electrode (type 4) without an ion exchanger exhibited an insignificant and unstable response to the HA solution because of partitioning ion exchange with ions of opposite charge (including primary ions) from solutions into the sensing membrane [35]. Due to a higher amount of ionophore, the electrode processing type 2 membrane provides increased sensitivity to HA and a lower detection limit of 8.58 × 10^−6^ M. However, the obvious improvement of another electrode (type 3) is not obtained in comparison with type 2. The similar performance of these two electrodes (type 2 and type 3) is mainly due to the insolubility of excessive Fe(TPP)Cl (6.5%, *w*/*w*) in the solvent.

According to the results, an ionophore content of 4.0% was selected for further investigation. To obtain the optimum thickness for an ion-selective membrane, three types of electrodes with different dipping times were fabricated, and the characteristics are shown in Table 2.

Type ii, which was dipped in the ion-selective membrane cocktail five times, provided relatively higher sensitivity and a lower detection limit. Additionally, the response time of type iii was more than 10 s because an increased thickness hinders ion transport. Due to the similar performance of types i and ii, the long-term stabilities were tested for optimum conditions. These two electrodes were used to detect 10^−7^–10^−1^ M HA solutions once every five days. After each use, the sensors were rinsed with deionized water and stored in a drying closet at room temperature. As shown in Figure 2A, type i worked normally for 15 days, and then, the sensitivity and reproducibility started to degrade. However, type ii could normally perform for at least 35 days, as shown in Figure 2B. Thus, the HA electrode dipped in the membrane cocktail five times could provide better characteristics, such as sensitivity, response time and long-term stability. Then, the electrode with an ionophore content of 4.0% *w*/*w* and dipped in the membrane cocktail five times was evaluated in the following part.

### 3.2. Calibration Curve and Stability of the Histamine Electrode

To achieve the diluted HA solutions with concentrations ranging from 10^−7^ to 10^−2^ M at pH 7.4, a certain volume of 0.1 M HA stock solution was added into 45 mL of acetic buffer (CH_3_COOH-LiOH) every 50 s with continuous magnetic stirring (400 rpm). Figure 3A shows the stepwise response after each addition, and the sensor reached a steady state in 5 s. The fast response time is attributed to the charge transfer at the interface of the selective membrane and solution. The experiment was repeated three times, and the calibration curve is plotted in Figure 3B.

The electrode showed a linear response in the range of 3 × 10^−5^–10^−2^ M, with a sensitivity of 46.42 mV/decade. The slope is slightly lower than the result derived in the Nernst equation, due to the minor forms of dicationic and neutral HA when it becomes protonated at an aliphatic amino group at neutral pH [36]. According to the IUPAC recommendations, the crosslink of the nonresponsive range and the Nernst equation in the calibration curve indicates that the detection limit is 8.58 × 10^−6^ M. The cation-π bond as well as the proper bond angle between metalloporphyrins and the imidazole ring in HA contributes to good sensitivity [32,37].

The stability was also examined by immersing the sensor into a 3 × 10^−4^ M HA solution for 15 min, during which the maximal fluctuation of potential value was ±0.65 mV. The relatively stable response over 15 min demonstrated that correction was not necessary in the examination.

### 3.3. Selectivity Coefficients

The selectivity coefficients for the HA sensor for the inorganic and organic ions existing in human plasma, including K^+^, Na^+^, Ca^2+^, GC, AA, HI, EP and Ur, were detected by FIM referring to the IUPAC recommendations. In this process, the potentiometric response to HA solutions from 10^−7^ to 10^−1^ M was determined against constant ionic backgrounds for KCl, NaCl, CaCl_2_, GC, AA, HI, EP and Ur, respectively. The logs of the selectivity coefficients were calculated and are shown in Table 3. The selectivity coefficients less than −1 indicate that the sensor was more sensitive to HA than other interferences. Compared with those for the other ions, the selectivity coefficients for K^+^ and EP were relatively high. However, the concentrations of these two ions in human plasma (K^+^: less than 3 mM; EP: less than 540 nM) [38,39] were lower than the experimental settings, so they would hardly cause interference practically. In addition, because of the proper bond angle, metalloporphyrins coordinate more closely with the five-membered imidazole ring in HA by removing the steric interaction with the hydrogens in the neighboring carbons [32].

### 3.4. Effects of pH and Temperature

To investigate the impact of pH on the HA sensor, the potentiometric responses to 10^−4^ M, 10^−3^ M and 10^−2^ M HA solutions at different pHs were measured. Considering the physiological condition in the human body, the pH range of 7.0–8.0 was provided as the background. As shown in Figure 4A, the sensor performed stably in the measured HA solutions, which indicates that the HA sensor’s performance was independent of the pH in the range of 7.0–8.0.

The temperature effect was also tested. Referring to the human body temperature, Figure 4B provides the basically stable response of the sensor in 10^−4^–10^−1^ M HA solutions within the temperature range of 35–41 °C. The sensitivity of 50.08 ± 0.79 mV/decade implies that a change in temperature in a certain range could hardly affect the performance of the HA sensor.

### 3.5. Reproducibility and Repeatability

In the experiment, five sensors made in the same batch were selected to measure 0.1 mM, 1 mM and 10 mM HA standard solutions in acetic buffer (CH_3_COOH-LiOH). The measured mean concentration (Mean), standard deviation (S.D.) and relative standard deviation (R.S.D.) are shown in Table 4. The mean concentration close to the real concentration and the R.S.D. less than 10% reveal the satisfactory reproducibility of the HA sensors.

The repeatability of the HA sensors was also evaluated. Six sensors were selected randomly to detect the response in 10^−7^–10^−1^ M HA solutions. As shown in Figure 5, the sensitivity of 50.96 ± 1.41 mV/decade demonstrates that the repeatability between different HA sensors was acceptable.

### 3.6. HA Determination in Artificial Cerebrospinal Fluid

To investigate the effect of interference ions, the performance of the HA sensor in artificial cerebrospinal fluid was tested by the standard addition method. A 10^−4^ M standard HA solution was prepared with the background of artificial cerebrospinal fluid. A certain amount of 10^−2^ M HA solution was added to obtain the increasing concentrations of 2 × 10^−4^ M, 3 × 10^−4^ M and 4 × 10^−4^ M. After repeating the test three times, the predicted concentration of the HA solution was calculated to be 9.97 × 10^−5^ M. Compared with the real concentration of 10^−4^ M, the recovery rate of 99.7% demonstrated that the common interference ions, for example, K^+^ and Na^+^, could hardly affect the determination of HA. Referring to the value of 1000 μM HA per gram of tissue in tumors and more than 50 mg/100 mg HA in poisonous fish, the recovery rate in 10^−4^ HA solution implied the potential usage of the sensor to monitor HA in biological fluids and poisonous foods.

## 4. Conclusions

In this study, miniaturized solid-contact HA sensors with 0.5 mm-diameter gold wires were made and evaluated. PEDOT:PSS coated on the wires transduced ionic signals by electronic conductivity. In the selective membrane, Fe(TPP)Cl, acting as an ionophore; 2-NPOE, acting as a plasticizer; and KTFPB, acting as an ion exchanger, were embedded in a PVC matrix for HA sensitivity. The HA electrode worked linearly in the range of 3 × 10^−5^–10^−1^ M, with a detection limit of 8.58 × 10^−6^ M. pHs within 7.0–8.0 and temperatures within 35–41 °C could hardly influence the stability of the sensor. The reproducibility, repeatability, stability and selectivity were also found to be acceptable. In addition, the performance of the HA sensor in artificial cerebrospinal fluid was analyzed. The recovery rate close to 100% suggests possible application in clinical research and food safety control.

## Figures and Tables

**Figure 1 sensors-21-06658-f001:**
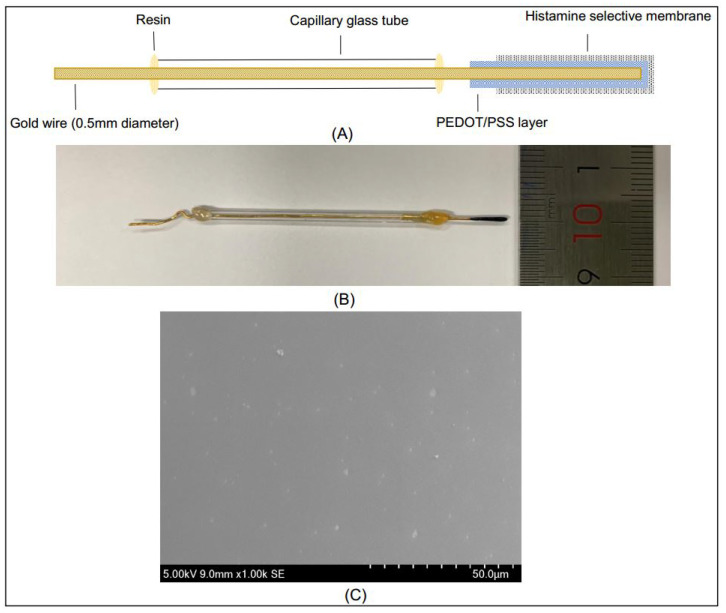
(**A**) Structure of solid-contact histamine ISE; (**B**) Photography of solid-contact histamine ISE (one centimeter ruler as a reference); (**C**) SEM image of histamine ion-selective membrane.

**Figure 2 sensors-21-06658-f002:**
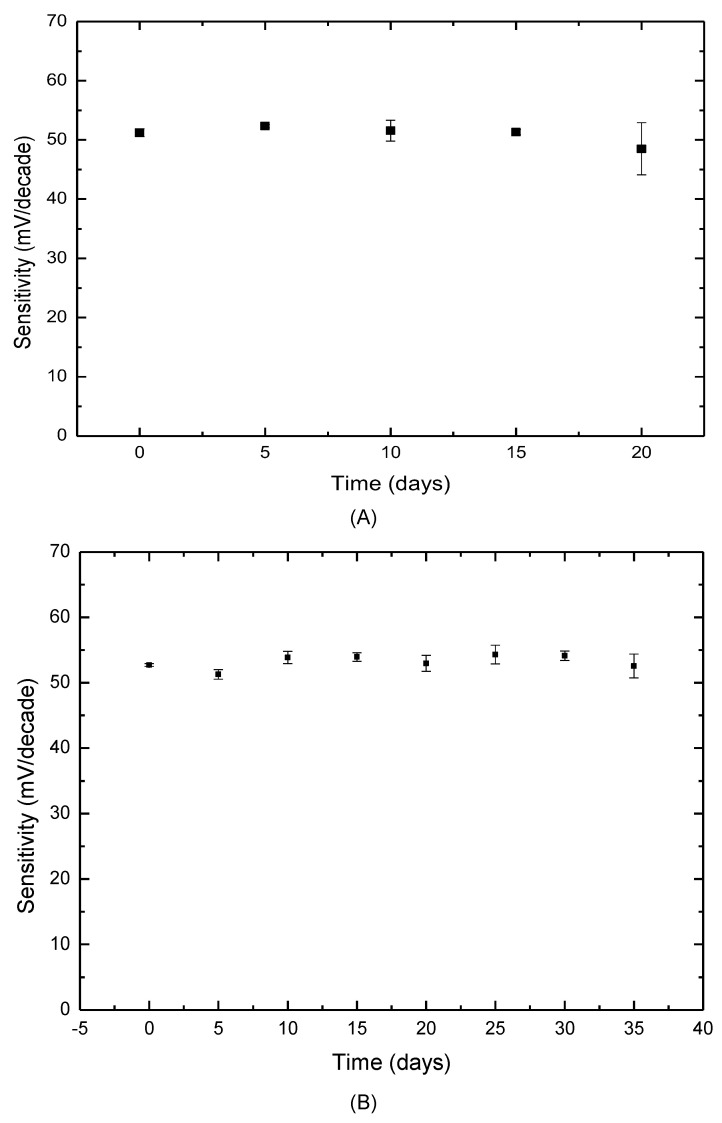
(**A**) Sensitivity of histamine electrodes (type i) over 20 days; (**B**) Sensitivity of histamine electrodes (type ii) over 35 days.

**Figure 3 sensors-21-06658-f003:**
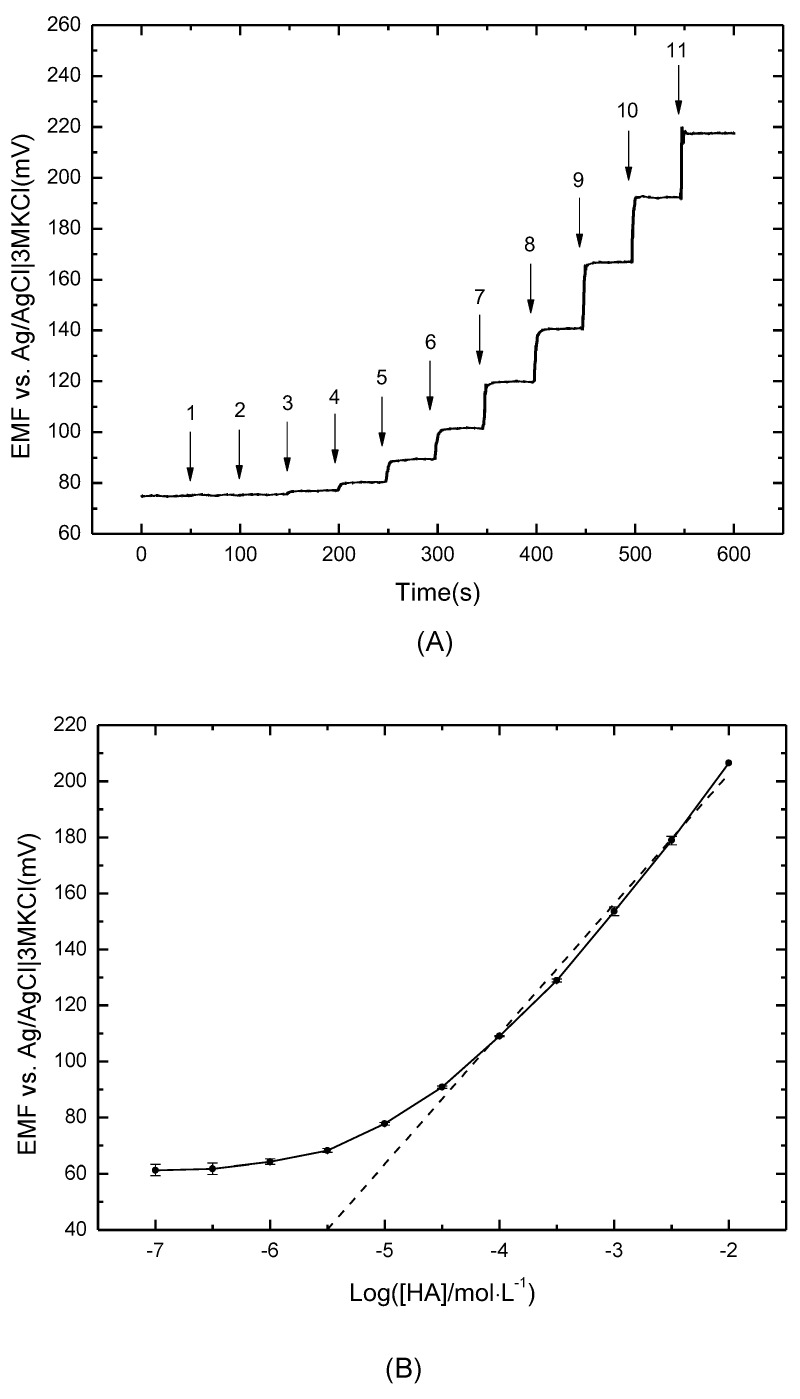
(**A**) Dynamic responses of histamine sensor with the addition of histamine stock solution every 50 s (1: 10^−7^ M; 2: 3 × 10^−7^ M; 3: 10^−6^ M; 4: 3 × 10^−6^ M; 5: 10^−5^ M; 6: 3 × 10^−5^ M; 7: 10^−4^ M; 8: 3 × 10^−4^ M; 9: 10^−3^ M; 10: 3 × 10^−3^ M; 11: 10^−2^ M); (**B**) Calibration curve of dynamic response in histamine solutions with the background of acetic buffer (CH3COOH-LiOH).

**Figure 4 sensors-21-06658-f004:**
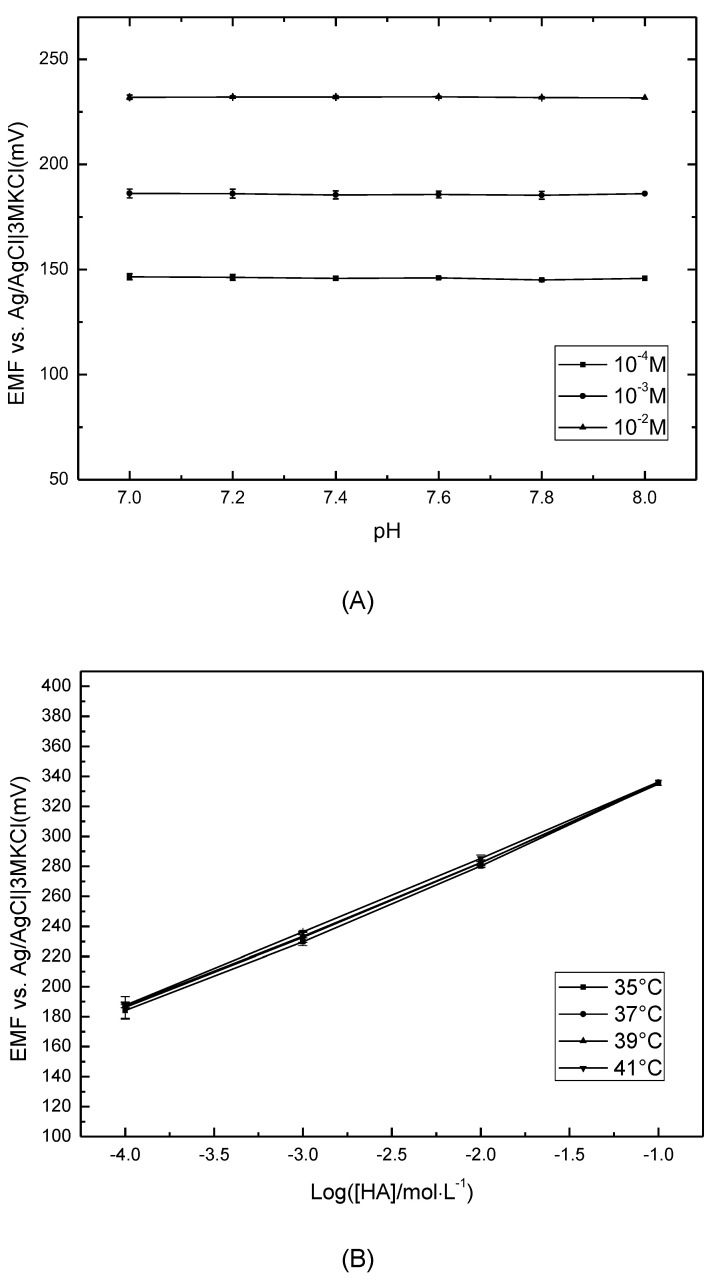
(**A**) Effect of pH on the potentiometric response to histamine solutions (10^−4^ M, 10^−3^ M and 10^−2^ M) with the background of acetic buffer (CH_3_COOH-LiOH); (**B**) Effect of temperature on the potential response to histamine solutions from 10^−4^ to 10^−1^ M.

**Figure 5 sensors-21-06658-f005:**
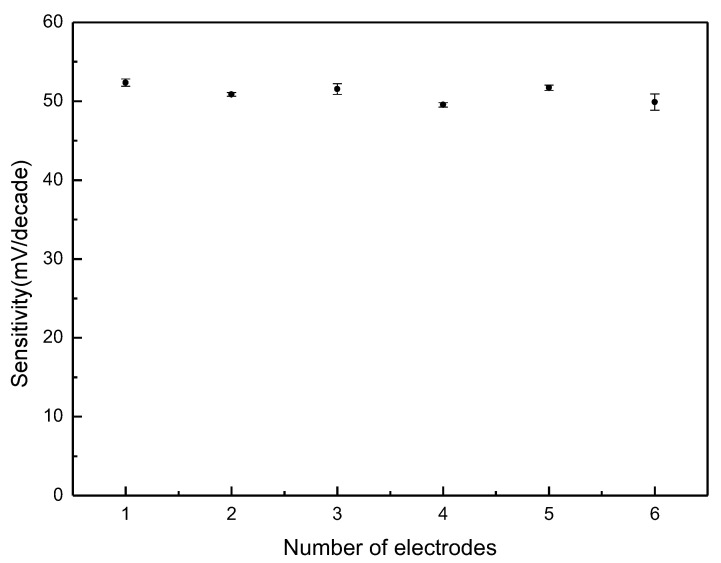
Repeatability of six histamine sensors in histamine solutions (10^−7^–10^−1^ M).

**Table 1 sensors-21-06658-t001:** Membrane composition and performance of histamine electrodes.

Type	KTFPB	Fe(TPP)Cl	PVC	2-NPOE	Sensitivity	Detection
	(% *w*/*w*)	(% *w*/*w*)	(% *w*/*w*)	(% *w*/*w*)	(mV/Decade)	Limit (M)
1	0.5	2.5	32.3	64.7	44.75	1.17 × 10^−5^
2	0.5	4.0	31.8	63.7	46.42	8.58 × 10^−6^
3	0.5	6.5	31.0	62.0	46.56	8.52 × 10^−6^
4	0	4.0	32.0	64.0	-	-

**Table 2 sensors-21-06658-t002:** Influence of dipping time on the characteristics of HA electrodes.

Type	Dipping	Sensitivity	Detection	Response	Long-Term
	Time	(mV/Decade)	Limit (M)	Time (s)	Stability (Days)
i	3	46.11	10^−5^	5	15
ii	5	46.42	8.58 × 10^−6^	5	35
iii	7	45.14	9.65 × 10^−6^	>10	-

**Table 3 sensors-21-06658-t003:** Selectivity coefficients for K^+^, Na^+^, Ca^2+^, GC, AA, HI, EP and Ur.

Interference *j*	Constant Concentration (M)	Selectivity Coefficient (Log ${}_{HA,j}^{Pot}$)
K${}^{+}$	0.01	−1.67
Na${}^{+}$	0.1	−3.10
Ca${}^{2+}$	0.1	−3.85
GC	0.1	−3.47
AA	0.01	−2.57
HI	0.05	−3.17
EP	0.01	−1.27
Ur	0.1	−3.50

**Table 4 sensors-21-06658-t004:** Reproducibility of five HA sensors.

Number of HA ISE		C(HA) (mM)		
		0.100	1.000	10.000
	Mean (mM)	0.101	1.008	9.455
No. 1	S.D. (mM)	0.002	0.021	0.209
	R.S.D. (%)	1.430	0.824	5.451
	Mean (mM)	0.099	1.030	10.069
No. 2	S.D. (mM)	0.002	0.017	0.764
	R.S.D. (%)	1.289	3.020	0.689
	Mean (mM)	0.101	1.051	10.536
No. 3	S.D. (mM)	0.004	0.088	0.823
	R.S.D. (%)	0.983	5.077	5.359
	Mean (mM)	0.107	0.995	10.306
No. 4	S.D. (mM)	0.002	0.026	0.217
	R.S.D. (%)	7.017	0.502	3.064
	Mean (mM)	0.109	1.020	10.964
No. 5	S.D. (mM)	0.004	0.037	0.393
	R.S.D. (%)	9.641	2.049	9.641

## Data Availability

The data presented in this study are available on request from the corresponding author. The data are not publicly available due to privacy.

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
