# Peer review of "Solid-Contact Ion-Selective Electrodes for Histamine Determination"

_sensors, 2021, doi:10.3390/s21196658_

Round 1
Reviewer 1 Report
Please see the attached file.

Reviewer 2 Report
Manuscript entittled: All-solid-state ion-selective electrodes for histamine concerns on a new potentiometric sensor based on porphyrine derivative. The authors showed very interesting study for histamine solid-contact analytical tool. In my opinion this manuscript needs major revision prior to publication proccess.
Below I have included comments and questions for the authors that should be considered prior to the publication of this manuscript.
- The authors already used the term ‘all-solid-state’ in the title of the manuscript. In my opinion this term is incorrectly used. All-solid-state potentiometric sensors are integrated analytical tools where reference and working electrodes have a the same phase (solid). See examplary works: https://doi.org/10.1016/j.snb.2014.06.067, https://doi.org/10.1002/elan.201700149. I propose to replace the term ‘all-solid-state’ with ‘solid-contact’.
- In the manuscript PEDOT/PSS abbreviation was used. This abbreviation suggest that PEDOT and PSS are two different layer. Doping should be written as PEDOT:PSS.
- Have the authors tried to perform electrosynthesis of the PEDOT:PSS layer ?If not, why did they use drop casting instead of electrosynthesis.
- Why Au electrodes were used instead of for example glassy carbon?
- For preparing HA standard solution using acetate buffer. It is not clear what the composition of this buffer is. I can only guess that the authors used lithium acetate and lithium hydroxide. If so, what is the influence of lithium ions on the potentiometric measurement (selectivity coefficient)?
- Why was gold dipped in PEDOT:PSS 18 times? How does the amount of immersion affect the sensor response?
- How the sensor response time was calculated?
- In Table 1 is (Log) and it should be (Log KHA/jpot).
- In Table 2 No. 4 was used twice.
- What was the sensitivity of the sensor? First, the authors wrote that it is equal to about 46 mV/dec. But, later is equal to about 50 mV/dec ? Why are there such differences.Has the same linear range has been used in the all calculations ?
- What was the detailed composition of the artificial cerebrospinal fluid ?
Authors should correct minor punctuation errors and add any missing spaces between the value and the unit.
Reviewer 3 Report
Manuscript ID: sensors-1317897
Title: All-solid-state ion-selective electrodes for histamine determination
The authors presented the use of solid-contact ion-selective electrodes for histamine (HA) determination.
The manuscript is of the Communication Type, but some aspects of the paper need to be improved:
The authors stated: "Compared with these ionophores, metalloporphyrins bond more stably with aromatic amines than aliphatic amines, which leads to the more strength interaction between porphyrins and HA in the membrane". Please explain the working principle of the proposed sensors and why these are solid-state ISEs.
In the section 2.3. the authors stated: "All experiments were done in 25â—¦C and pH value of 7.4, in which the monoprotonated form of HA is dominated." Please explain how this was obtained during the potentiometric measurements if in the Section 3.1 the authors stated: "In this experiment, to achieve the diluted HA solution with the concentration ranging from 10−7 to 10 −2M at pH 7.4, a certain volume of 0.1 M HA stock solution was added into the 45mL acetate buffer ..."? Because the pH of the acetate buffer is less than 5, this means that the pH of the final HA solution will be less than 7.00. This also means that Section 3.3 should be redone.
Section 3.2. Please detail how the selectivity coefficient were obtained (calculated).
Round 2
Reviewer 1 Report
Please see the attached file

Reviewer 2 Report
Thank for Your answers. The work can be published.
Author Response
Thanks for your careful work and professional suggestions. They are helpful to improve the quality of our manuscript.
Reviewer 3 Report
Solid-contact ion-selective electrodes for histamine determination
The authors explained and modified the unclear notions. Thus, I recommend the publication of the manuscript in the revised form.
Author Response
Thanks for your careful work and suggestions. They are helpful to improve the quality of our manuscript.